# Environmental Contamination by *Echinococcus* spp. Eggs as a Risk for Human Health in Educational Farms of Sardinia, Italy

**DOI:** 10.3390/vetsci9030143

**Published:** 2022-03-18

**Authors:** Elisa Serra, Gabriella Masu, Valentina Chisu, Stefano Cappai, Giovanna Masala, Federica Loi, Toni Piseddu

**Affiliations:** 1OIE-National Reference Laboratory of Echinococcosis, Istituto Zooprofilattico Sperimentale della Sardegna, Via Vienna, 07100 Sassari, Italy; elisa.serra@izs-sardegna.it (E.S.); gabriella.masu@izs-sardegna.it (G.M.); giovanna.masala@izs-sardegna.it (G.M.); toni.piseddu@izs-sardegna.it (T.P.); 2Osservatorio Epidemiologico Veterinario Regionale della Sardegna, Istituto Zooprofilattico Sperimentale della Sardegna, Via XX Settembre, 09125 Cagliari, Italy; stefano.cappai@izs-sardegna.it

**Keywords:** *Echinococcus granulosus*, educational farm, environmental contamination, one health approach

## Abstract

Cystic Echinococcosis (CE) is a severe zoonosis caused by the larval stage of the tapeworm *Echinococcus granulosus*. These parasites are naturally transmitted between canid definitive hosts that harbour the adult stage in the intestine, and mainly ungulate intermediate hosts, with larval cysts developing in their internal organs. The close coexistence between dog and sheep with incorrect hygiene management are the most important factors for the persistence of this parasitic pathology. The Educational Farms (EF) are farms and agritourisms suitably equipped to carry out training activities for people interested in farm practices and agricultural processing, in particular student groups. Public attendance of farms represents a new potential risk factor for the zoonoses transmission. Consumption of contaminated food and water in combination with contact or playing with domestic dogs (*Canis familiaris*) are possible routes of zoonoses human infection. In fact, *Echinococcus* spp. eggs may persist in the environment up to several months at low temperatures and moist conditions, having the chance of contaminating different matrices and surfaces. The aim of this investigation was to study environmental contamination by parasitic elements as a risk for zoonoses, such as *Echinococcus* spp. A total of 116 samples (35 of water, 33 of soil, 23 of vegetables, 25 of dog faeces) were collected in 30 EF in Sardinia. Samples were subjected to biomolecular investigation for the research of specific gene sequences of *Echinococcus granulosus*, *Echinococcus multilocularis* and *Taenia* spp. The study allowed to identify eight positive EF due to the presence of *Echinococcus*
*granulosus* in eight dog faeces samples and one positive EF due to the presence of *Taenia* spp. in a water sample. The work has allowed to develop and harmonise the diagnostic methods and operating protocols essential for controlling the spread of the CE to create “One Health” intervention plans in high endemic areas through the implementation of SOP (standard operating procedures) for monitoring the pathology in animals, humans and environment.

## 1. Introduction

Cystic Echinococcosis (CE) is a chronic and disabling neglected zoonotic infection caused by larval stage of taeniid tapeworm *Echinococcus granulosus* (*E.g.*) sensu lato, a species complex that includes several genotypes and cryptic species [1]. The current view, informed by epidemiology, biology and in particular molecular genotyping, recommends the inclusion of five species: *E.g.* sensu stricto (G1–G3), *E. equinus* (G4), *E. ortleppi* (G5), *E. canadensis* (G6/G7, G8 and G10) and *E. felidis* [2,3]. *E.g.* G1 is the most widespread genotype in human CE cases (88.4%) [4].

The tapeworm lifecycle includes dogs and other canids as definitive hosts and numerous ungulate (as sheep, cattle, pig, and goat) as intermediate hosts. Definitive hosts are infected through ingestion of parasite larvae, by feeding on the infected offal of livestock or hunting [5]. In humans, defined as accidental intermediate host, after oral ingestion of *E.g.* eggs consequently to hand-to-mouth contact with contaminated matrices (i.e., dog faeces, dog fur, soil, food, water) cysts may develop in many anatomic sites, particularly liver and lungs, [6]. Identifying the time of the infestation of this parasitosis is particularly complex due to the long incubation period and faecal-oral transmission cycle from oral egg intake, through contact with dogs or with contaminated soil, water and food [7]. The number clinically diagnosed cases is only a part of the total number of infected individuals. This leads to an underestimation of the presence of the disease worldwide, confirming its classification as “Neglected Tropical Disease” by the World Health Organisation (WHO). Despite the *E.g.* is a cosmopolitan species, its presence is more common in the rural areas of southern and eastern Europe, central Asia and South America [8,9,10]. Environmental and anthropogenic factors influence the CE distribution in different parts of the world [10]. The highest prevalence among humans and animals occurs where several risk factors are still in place: extensive livestock production, keeping a large numbers of dogs with free access (or fed with) to raw viscera of animals slaughtered without meat inspection, inadequate systems for the slaughtering by products disposal, lack of anthelminthic treatment of dogs and populations, poor health education [11,12].

Recently, the number of livestock farms embarking the way of “multifunctionality” as a business strategy is increasing by transforming farm products in mini dairies or small farm delicatessen and opening the company to visitors and to school groups. Educational farms (EF) are farms and part of agritourism that are suitably equipped to carry out training activities for people interested in rural lifestyle, in particular students and organised groups. The activities are led by specialised personnel and designed to make both adults and children discover the typical life in the farm. Children will be in close contact with the animals, learning their characteristics and ethology. This is the new face of rural tourism, a way to rediscover regional traditions and to better know the countryside work. An important contribution of hand-to-mouth transmission to the possibility of acquiring CE seems more plausible in children than in adults due to their behaviours, particularly the frequency of hand-to-mouth contacts. This is a fundamental variable to the estimation of the CE exposure assessment. In fact, several studies have shown that children have an even greater potential for exposure to chemicals available through the non-food route of ingestion [13]. Attending farms for educational and recreational purposes may represent a new potential risk context for the transmission of zoonoses. The *E.g.* eggs have a strong resistance to adverse environmental conditions: they can survive in the environment with very wide temperature tolerance limits (from +40 °C to −70 °C) and resistance in moist soil and water for up to 16 months, resulting in excellent biological risk indicators [14]. In this context, people could become infected by drinking contaminated water or coming in contact with contaminated soil, food and dogs that may be a source of human infection.

In order to have a deeper insight into the source of *E.g.* human contamination, this is the first report in which a representative number of EF from different geographical Sardinian areas were evaluated in order to detect the presence of *E.g.* in soil, water, vegetables and dog faeces and assessing the risk factor associated with the infection in humans.

## 2. Materials and Methods

### 2.1. Study Area and Contest Analysis

This study was carried out in Sardinia, the second largest island in the Mediterranean Sea, with a surface area of 24,089 Km^2^. It is the fourth least populated region in Italy with 1,611,621 inhabitants in 2020, having a low mean population density of 69 inhabitants/Km^2^ compared to 200 inhabitants/Km^2^ in Italy. The larger urban centres are located near the coastline, while the inner area is sparsely populated.

Currently, the Sardinian sheep farms are 14,875 accounting for about half of the Italian stock registering a total of 3,063,077 sheep. The animal density peaks to 125,75/Km^2^ compared to 21,66/Km^2^ in the Italian mainland (National Italian Database 2020 (BDN), established by Ministry of Health at the National Surveillance Centre of the IZS in the Abruzzo and Molise Region). The island is divided into eight Local Health Unit (Azienda Sanitaria Locale—ASL), covering most of the territory. To obtain an adequate sample distribution we randomly sampled the EF based on their spatial distribution, and taking into account the number of animal bred.

### 2.2. Recruitment of the Education Farms

During 2018, a total number of 100 EF located throughout Sardinian Region were contacted via phone and invited to participate in this study. An anonymous form was obtained from each EF to gather data here presented. Based on an expected prevalence of contaminated soil, water, vegetables and dog faeces collected from farms of 3% with a power of 80% (α = 0.05, two-sided), the sample size necessary to estimate a true prevalence and the risk factors for human was calculated to be of 30 farms.

For each educational farm involved in this study the following factors were evaluated: heritage livestock, presence or absence of vegetable garden, garden fence, distinction between vegetable and educational garden, type of water used for the irrigation, presence of dogs and type of educational path proposed to visitors (animal husbandry, crop health, protection of landscape, nature conservation and agricultural production).

### 2.3. Samples Collections

From January 2018 to April 2019, 116 specimens (water, soil, vegetables and dog faeces) collected from 30 different Sardinian farms were analysed for the molecular detection of cestode eggs, as shown in Table 1.

According to the protocol described by Collender et al. in 2015 [15], soil samples could be collected from three different areas of an EF:Reception area used to greet the guests and to provide them a waiting area;Educational path equipped to carry out training activities for people interested to go deeper into rural areas;Areas used for recreational activities.

Samples of the water used for the irrigation washing of vegetables or drinking were obtained from each farm (i.e., from nearby sources, cistern, public supply or stream) and analysed for the presence of cestode contamination. Samples were stored at a temperature 4 °C and sent to the laboratory of CeNRE, and processed within 24 h of collection.

Vegetable samples, mainly broad-leaved, were randomly collected from both productive and educational gardens. Faecal samples were collected from the environment and classified in two groups depending on whether or not they could be associated with dogs. Samples were kept at a temperature of 4–8 °C and sent to the laboratory of IZS and processed within 24 h after collection. Samples were stored for at least 3 days at −80 °C to kill the parasitic eggs of *E.g.* [16] and then transferred to a temperature of −20 °C, until further analysis. Samples were subjected to a combination of sedimentation, flotation and filtration by using sieves with mesh of different sizes, as suggested by Mathis et al. in 1996 [17].

### 2.4. Sample Processing

The isolation of cestode eggs from the collected samples was performed by different methods depending on the different matrices.

#### 2.4.1. Soil

A sieve of 4 mm^2^ mesh was used to remove stones and large pieces of organic matter from soil (400 g). Tween−80 solution (60 mL; 0.05%) was added to sieved soil (40 g) and mixed for 20 min. Obtained samples were divided equally into 30 mL tubes and centrifuged at 220× *g* for 3 min. The supernatant was removed, and the pellet was washed with distilled water, vortexed and centrifuged at 220× *g* for 3 min. Pellet was resuspended in 30 mL of zinc chloride (ZnCl_2_) solution, vortexed and centrifuged at 220× *g* for 3 min. The ZnCl_2_ solution was then added to form a convex meniscus and a 20 × 20 mm coverslip was placed on the edge of the tube for 15 min. The slide was then washed with distilled water and the obtained suspension was transferred to a 2 mL centrifuge tube (Eppendorf Mississauga, Ontario, Canada), vortexed and centrifuged at 300× *g* for 1 min. The supernatant was collected, and the pellet was stored at −80 °C until DNA extraction [15].

#### 2.4.2. Water

Water samples (1 L) were filtered, as previously described [18]. Briefly, the filtration was done using 8 µm cellulose filters by vacuum pump. The filter was then transferred to a 50 mL tube containing 0.2% solution of Tween−20. The filter was discarded, and the suspension was centrifuged at 220× *g* for 5 min. The supernatant was removed and the pellet frozen at −80 °C until DNA extraction.

#### 2.4.3. Vegetables

Vegetable samples (350–500 g) were soaked in water. The obtained suspension was filtered with a polyethylene funnel (diameter 30 cm). Vegetables were further rinsed twice and the collected liquid was filtered by using a 100 µm metal filter and then a 50µm filter. The 50µm filter was washed with Tween−20 solution (0.2%) and the washing liquid was subsequently filtered through of a vacuum pump using a 8µm cellulose membrane filters into 50 mL tubes. The filtered liquid was vortexed and centrifuged at 220× *g* for 5 min. The supernatant was removed, and the pellet was frozen at −80 °C until DNA extraction [19,20].

#### 2.4.4. Faecal Samples

Flotation solution (10 mL of Sheather’s solution: 454 g granulated sugar, 355 mL tap water, 6 mL formaldehyde 4%; gravity 1.27) was added to 5 g of faecal samples (5 g) as previously described [21]. Briefly, the filtration was performed using a tightly meshed kitchen strainer in a 500 mL glass conical and graduated. The filter was then transferred into a 15 mL tube and centrifuged at 26× *g* for 5 min. Floating solution was then added to form a convex meniscus on a 20 × 20 mm coverslip. Coverslip was then observed according to a “Greek decoration” under optical microscope (Axioplan, Carl-Zeiss-Strasse, Oberkochen, Germany) using 10× magnification. If the sample was positive, physiological solution (4–5 mL) was added to 2 g of faeces and then filtered using a 100 μm cell strainer. The filter was transferred to a 50 mL tube containing 1.5 mL of Percoll and 40 mL of 0.01 M PBS and centrifuged at 232× *g* for 30 min. The supernatant was discarded and the pellet was stored at −80 °C until DNA extraction [17].

### 2.5. Molecular Detection and Characterisation of Cestode Eggs

The “QIAamp DNA Stool Mini Kit” (Qiagen, Hilden, Germany) was used for DNA extraction from matrices as above, according to the manufacturer’s instructions with minor modifications as follows: 50 μL of Chelex beads (50% *w*/*v* in distilled water; Bio-Rad Laboratories, Inc., Hercules, California, CA, USA) was added to enhance the capture of DNA from cestode eggs. After homogenisation by using Tissue Lyser II (Qiagen, Hilden, Germany) for 30 min. at room temperature, the procedure followed the manufacturer’s instructions of the kit. The extracted DNA was stored at −80 °C until further analysis. DNA extracted was then subjected to polymerase chain reactions (PCR) to detect the DNA of *Echinococcus* spp. from different matrices.

#### 2.5.1. Multiplex PCR for Discrimination of *E.g.* Sensu Lato and *Echinococcus multilocularis* from Other Cestodes

To detect *E.g.*, *Echinococcus multilocularis* (*E.m.*) and other cestodes from extracted samples, a multiplex PCR assay was used as previously described by Trachsel et al. in 2007 [22] by using three pairs of primers listed in Table 2.

The PCR reaction mixture was carried out in a final volume of 25 µL including 12.5 µL QuantiTect Probe PCR Master Mix (Qiagen, Toronto, Canada) (1× final concentration), Milli-Q water RNAse-free (9 µL), 2.5 µL each forward and reverse primer (2 µM) and 1 µL DNA template. The amplification program consisted of an initial denaturation step of 15 min at 95 °C, followed by 40 cycles of 30 s at 94 °C, 90 sec annealing at 58 °C and 10 sec at 72 °C, and 1 cycle of 5 min at 72 °C. Amplicon products were electrophoretically separated in 2% agarose gel under standard conditions. DNA Molecular Weight Marker VIII (Roche, Basilea, Switzerland) was used for DNA sizing. The products were treated with nontoxic SYBR^®^ Green DNA Gel Stain (Invitrogen, Carlsbad, CA, USA), and visualised using standard UV transillumination.

#### 2.5.2. CoproPCR Assay for *E. granulosus* s.l. Detection

DNA from faecal samples was amplified by classical PCR using the protocol described by Lett et al., in 2018 [23], including the primers Eg1121a [5′-GAA TGC AAG CAG CAG ATG-3′] and Eg1122a [5′-GAG ATG AGT GAG AAG GAG TG-3′] targeting a fragment of 133 bp of the repeated unit EgG1 HaeIII. The amplification reaction mixture (50 µL) consisted of 5 µL of buffer PCR 10X with MgCl_2_ (15 mM), 10 µL of dNTPs (1.25 mM), 2 µL of Eg1121a (25 pmol/μL), 2 µL of Eg1122a (25 pmol/μL), 0.5 µL of Ampli Taq Gold (5U/μL), 28.5 µL of Milli-Q water RNAse-free, 1 µL Hi-Di Formamide and 1 µL of template DNA. PCR reactions were performed with a Ampli Taq thermocycler (Thermo Fisher Scientific Inc., Waltham, MA, USA) and consisted of an initial step at 95 °C for 5 min, followed by 35 cycles of 60 sec at 95 °C, 30 sec at 55 °C, 60 sec at 72 °C and a final extension step at 72 °C for 10 min. DNA amplicons were separated on 1.5% agarose gel and bands were visualised by using SYBR^®^ Safe DNA gel stain (Thermo Fisher Scientific Inc., Waltham, MA, USA).

### 2.6. Data Processing

Details about the EF context were recorded and evaluated based on their macro and micro context. First, the macro area (ASL) was evaluated based on sheep farms and animal density (i.e., sheep and goats) by Km^2^, and the raw number of sheep. To this aim, data from the Veterinary Information Systems of the Italian Ministry of Health (VETINFO) and National Zootechnical Database (BDN) were collected.

Furthermore, to study in deep the epidemiological context within the area surrounding, each EF was examined to identify the concomitant presence of sheep farms within a radius of 2 Km (average of pasture range in Sardinia) [24]. After georeferencing each farm with latitude and longitude, the buffering function [25] was implemented in ArcGIS software (version 10.2, ESRI, Redlands, CA, USA) to create the 2 Km buffer around defining the reference area. Indeed, to allow the evaluation of the territorial framework and to study the potential cross-contamination with nearby sheep farms, the number of dogs present in the farm and the presence/absence of fences were recorded during the on-field inspections.

Data about farm information such as farm code, farm owner’s name and surname, unique personal ID (fiscal code), address, location, municipality, province, ASL of reference, latitude and longitude, business start date, business end date and animal census data were collected in specific database. All confidential information was encrypted before analysis to guarantee the privacy of the farm owners. All information collected was password protected to ensure data security.

Statistical descriptive analyses were performed using Stata software (StataCorp. 2013. STATA Statistical Software, release13, StataCorp LP, College Station, TX, USA). Quantitative variables are summarised as mean and standard deviation (SD), median and interquartile range [I-III quartile]; qualitative variables were summarised as number and percentage (%). To compare qualitative variables, either the Chi-square test or the Fisher exact test was applied, while differences in quantitative variables were evaluated based on the Kruskal–Wallis nonparametric test. The level of *p* < 0.05 was considered significant.

## 3. Results

A total of 30 EF agreed to participate in this study. The EF were overall distributed within the 8 ASL based on the farm density and animal density, as reported in Table 3 and illustrated in Figure 1.

Eight faeces samples were positive to the of *E.granulosus* specific EgG1 HaeIII copro –PCR. One water sample tested positive to the presence of *Taenia* spp. by multiplex PCR (Table 4). The positive samples were isolated from 8 EF belonging to the ASL 4 area (Figure 1).

Concerning the variables considered on all the EF from this study, farms were divided into two groups as positive or negative for *E. granulosus* as summarise in Table 5.

From the study of EF characteristics, it emerged that positive and negative EF were on median located on 250 MASL [I-III quartile = 75–675] and 175 MASL [I-III quartile = 100–350], respectively (*p* = 0.52). Epidemiological context of positive EF was characterised by borderline lower distance from other farms (mean = 363, SD = 203) with respect to negatives (mean = 640, SD = 419) (*p* = 0.06) and a similar number of farms located within the 2 Km radius of positive (mean = 9.6, SD = 4.1) and negative farms (mean = 10.0, SD = 6.1) (*p* = 0.93). Most of the positive EFs were non-confined except three (20% of the overall fenced). In 24 over the 30 EFs, there were dogs in numbers ranging from two to seven, similarly distributed within positive and negative EFs (*p* = 0.78). Borderline differences (*p*-value = 0.1) were observed in the median number of animals breed between positive and negative EFs, particularly with an increasing number of sheep and goats in positive (median of sheep: 400 [185–412], median of goats: 20 [3–50]) and negative EFs (median of sheep: 100 [43–120], median of goats: 5 [1–40]).

## 4. Discussion

The management of sheep farming in Sardinia is semi-extensive with minimal use of fences. Therefore, sheep are free to graze in land where there is the possibility of other domestic and wild animals passing through. Such a context in combination with the difficult management of dogs in rural environments, facilitate the completion of the parasitic cycle. Recent studies regarding potential risk factors confirm that the presence of stray dogs, their free access to carcasses of intermediate hosts not removed from the fields and the keeping of dogs in poorly managed rural contexts, are statistically significant elements associated with the perpetuation of the parasite cycle in endemic areas [26]. In Sardinia dogs are raised as pets, for protection, guarding or hunting and, especially in rural areas, as shepherd dogs that are often free to roam and without an adequate parasiticide treatment. Currently, despite the existence of a regional canine registry, data about farm dogs are unknown. Based on information collected during the project’s management, total number of shepherd dogs was 104 with a median number of 3 dogs per farm (min: 0, max: 16). This data confirms and updates a study previously conducted by CeNRE on 700 farms in 42 Sardinian municipalities (unpublished data) and the previous study of Varcasia et al. in 2011 [27] in which the total number of shepherd dogs in the 172 farms was 652, an average of 3.8 dogs per farm. The farmers keep several dogs for guarding livestock, free to roam. The free dogs cover several kilometres and may become infected by feeding on carcasses of infested intermediate hosts abandoned in the fields. Subsequently, they may contaminate the soil with *E.g.* eggs potentially present in their faeces. Moreover, sheep plays an important role as intermediate host in widespread contamination of *E.g.* in the territory. Thus, sheep constitute a potential reservoir of disease having a long productive life of about 6 years. This study highlights that positive farm are those in which a high density of farms around the 2 km radius, and higher number of animals bred (particularly sheep and goats) are present, even if these differences are not statistically significant. Furthermore, the standard hygiene measures in Sardinian farms are often poor, and low attention in removing the carcasses from the fields as well as the wrong habits of slaughterhouse workers should be taken into account [28].

Determining the prevalence at which eggs are shed into the environment and their capacity to survive is fundamental to ascertain the real endemic status of CE in an area [29]. For the context analysis it is necessary to define the prevalence of human CE. CeNRE conducted several studies to analyse data from the hospital discharge records (HDRs) with CE-related diagnoses provided by the Ministry of Health of Italy. Brundu et al. in 2014 [30] analysed a total of 16,550 HDRs from 2001 to 2012 related to the admission of 10,682 Italian patients. The HDRs were analysed according to the patient’s region and province code to evaluate the demographic and clinical characteristics of each case, together with the annual incidence rates of hospital cases (AIh) in administrative divisions in rural and urban areas. The highest average AIh was registered in Sardinia 6.9/105, followed by the South of Italy with an average AIh of 1.9/105 inhabitants (5.4/105 inhabitants in Basilicata) and the Centre with an average AIh of 1.07/105 inhabitants (1.65/105 in Latium). The assessment of economic remuneration by Italian Regions for HDRs from 2001 to 2014 shows that in Sardinia CE causes serious global human health concerns and leads to significant economic losses arising from the costs of medical treatment, morbidity, life impairments and fatality rates in human cases. Sardinia spent €4,523,600 and the national annual average direct cost for 100,000 inhabitants was €6398 and in Sardinia was €19,523.77 [31]. Human and animal CE cases are strictly associated with the presence of livestock farms [26]. In fact, previous studies [32,33] demonstrated infections depending on variables (i.e., personal hygiene, socio-economic contest) that facilitate close contact with *E.g.* eggs [34]. The farms recently provide leisure services based on outdoor activities increasingly complementing quality food production and some have developed teaching and learning activities for school children [35]. Therefore, this scenario represents a new potential risk factor for the zoonoses transmission. Small children are at risk from geohelminth infections because of their lifestyle and their playing environment [36,37]. Several studies recognise public parks, particularly playgrounds that are heavily fouled by dogs and cats, as an important source of infection [37]. Therefore, soil-to-hand-to-mouth could be an important transmission route within areas where *E.g.*-infected dogs are kept, for both dog owners and the neighbouring community [13].

## 5. Conclusions

This study confirms that the combination dog-sheep is a key element for the persistence of the CE cycle and highlights the extreme complexity of geographically exactly identifying the source of the disease. Therefore, in planning CE control actions in highly endemic and extended areas, it is necessary to provide interventions aimed at containing the phenomenon of “canine vagantism”. On the other hand, setting up a careful surveillance of the intermediate host during the slaughter is a key point. In fact, although EF are known to be “virtuous” in term of hygiene management, it emerges that “canine vagantism” can affect the spread of this zoonosis, through environmental contamination, and thus favouring the infestation of sheep and the perpetuation of the biological cycle of *E.g.* Moreover, the role of Public Veterinary Services (ASL), which is a key tool in preventing the disease spread, is even more crucial, as well as the implementation of specific dedicated programs and awareness.

## Figures and Tables

**Figure 1 vetsci-09-00143-f001:**
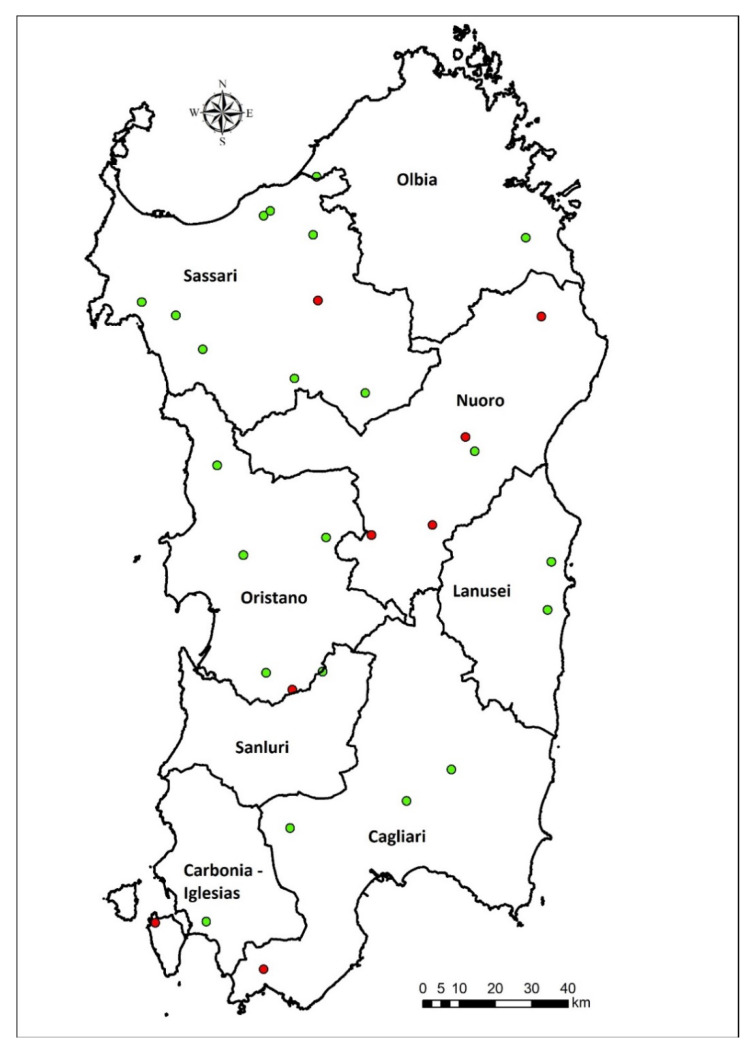
Map of the educational farms sampled. Positive samples are represented by red points, while negative samples are highlighted with green points. ASL limits are represented by black lines.

**Table 1 vetsci-09-00143-t001:** Samples distribution by matrices and place.

Matrices	Sample Group	Frequency (Number, %)
Water	Well	15 (43%)
	Cistern	8 (24%)
	Public supply	10 (28%)
	Stream	2 (5%)
Total		35 (30%)
Soil	Reception point	12 (36%)
	Educational path	16 (49%)
	Recreational point	5 (15%)
Total		33 (28%)
Vegetables	Production garden	18 (78%)
	Educational garden	5 (22%)
Total		23 (20%)
Dog Faeces	Identified dogs	8 (32%)
	Unidentified dogs	17 (68%)
Total		25 (22%)
Overall samples		116

**Table 2 vetsci-09-00143-t002:** Primers used in this study for amplification of gene sequences specific for *E. granulosus*, *E. multilocularis* and *Taenia* spp.

Parasite	Gene Target	Primers Pairs	Primer Sequences	Amplicon Size
*E. multilocularis*	NADH dehydrogenase subunit 1 (nad1)	Cest1	5′-TGCTGATTTGTTAAAGTTAGTGATC-3′	395 bp
		Cest2	5′-CATAAATCAATGGAAACAACAACAAG-3′	
*E. granulosus*	Ribosomal RNA (rrnS) subunit	Cest4	5′-GTTTTTGTGTGTTACATTAATAAGGGTG-3′	117 bp
		Cest5	GCGGTGTGTACMTGAGCTAAAC-3′	
*Taenia* spp.	Ribosomal RNA (rrnS) subunit	Cest3	5′-YGAYTCTTTTTAGGGGAAGGTGTG-3′	267 bp
		Cest5	5′-GCGGTGTGTACMTGAGCTAAAC-3′	

**Table 3 vetsci-09-00143-t003:** Analysis of territorial context for each ASL: sheep farms per Km^2^, sheep per Km^2^, total number of sheep. Data are presented as number (percentage, %) or mean and standard deviation (SD), calculated by municipalities included in each ASL.

ASL	No. of Sheep Farms	Sheep Farms Density by Km^2^	Sheep Density by Km^2^	No. of Sheep	EF Sampled
01-Sassari	3821 (20.6%)	0.77 (0.22)	189 (75)	803,695 (20.6%)	9 (30%)
02-Olbia	1513 (8.1%)	0.27 (0.41)	41 (19)	146,504 (4.8%)	2 (7%)
03-Nuoro	4400 (23.7%)	0.95 (0.19)	191 (78)	765,678 (25.0%)	6 (20%)
04-Lanusei	1218 (6.6%)	0.84 (0.21)	39 (27)	73,198 (2.4%)	2 (7%)
05-Oristano	2602 (14.0%)	0.81 (0.15)	158 (69)	485,477 (15.9%)	5 (16%)
06-Sanluri	1134 (6.1%)	0.58 (0.26)	139 (78)	213,028 (7.0%)	1 (3%)
07-Carbonia-Iglesias	1170 (6.3%)	0.72 (0.33)	110 (51)	140,612 (4.6%)	2 (7%)
08-Cagliari	2729 (14.7%)	0.83 (0.17)	116 (58)	429,930 (14.1%)	4 (13%)
Total	18587	0.77 (0.23)	126 (70)	3,058,122	30

**Table 4 vetsci-09-00143-t004:** Samples collected from the 30 educational farms involved in this study.

Soil (*n* = 33)			Water (*n* = 35)				Vegetables (*n* = 23)		Faeces (*n* = 25)	
RR	EP	RA	W	C	PS	S	PG	EG	ID	UD
12 Neg	16 Neg	5 Neg	15 Neg	1 Pos *Taenia* spp. 7 Neg	10 Neg	2 Neg	18 Neg	5 Neg	2 Pos *E.g.* 6 Neg	6 Pos *E.g.* 11 Neg

RR: reception place; EP: educational path; RA: recreational activities; W: well; C: cistern; PS: public supply; S: stream; PG: productive garden; EG: educational garden; ID: identified dog; UD: unidentified dog.

**Table 5 vetsci-09-00143-t005:** Distribution of the baseline variables characterizing the 30 educational farms involved. Data are presented as mean (SD), or number of EF and median [I-III quartile], or number of EF (percentage, %).

Variables	Positive EF (*n* = 8)	Negative EF (*n* = 22)	Overall (*n* = 30)
Altimetry (MASL)	250 [75–675]	175 [100–350]	200 [100–400]
Distance from farms (m)	363 (203)	640 (419)	566 (390)
No. farms located in 2 km of radius	9.6 (4.1)	10.0 (6.1)	9.9 (5.5)
Fenced	3 (20%)	12 (80%)	15 (50%)
Species			
Dogs	8, 4 [3–5]	16, 3 [0–5]	24, 3 [1–5]
Cats	4, 4 [2–5]	11, 3 [3–6]	15, 3 [3–6]
Sheep	5, 400 [185–412]	9, 100 [43–120]	14, 110 [43–412]
Goats	3, 20 [3–50]	5, 5 [1–40]	8, 12 [2–45]
Pigs	4, 15 [6–34]	10, 8 [6–10]	14, 8 [6–24]
Poultry	4, 8 [4–12]	7, 15 [8–50]	11, 10 [6–20]
Horses	4, 4 [3–6]	4, 1 [1–4]	8, 3 [2–5]
Donkeys/Mules	2, 2 [2–2]	3, 3 [2–4]	5, 2 [2,3]
Other (cattle, deer, muflon, wild boar)	2, 15 [4–26]	8, 6 [5–16]	10, 6 [5–20]

## Data Availability

The data presented in this study are available in the manuscript.

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
