# Peer review of "Environmental Contamination by Echinococcus spp. Eggs as a Risk for Human Health in Educational Farms of Sardinia, Italy"

_vetsci, 2022, doi:10.3390/vetsci9030143_

Round 1

Reviewer 1 Report

LINE 44- E. equinus in latin

Line 121 - erase "clinical" specimens is enough

Line 180- do not forget to consider the ocular magnification (8 or 10 ?)

Figure 1 (map) - Olbia: where is the secon point?

Line 343- reference number is 38

Line 343 - the Italian experiences in this specific topic are lacking

Line 347 - remove  the first sentence

To add in conclusions : More on the implementation of the role of Public Veterinary Services (AUSL) with dedicated programs on this specific context.

Author Response

Dear review,

thank you very much for your valid and useful revision. Here as follow the point by point assessment.

Rev.1

Dear Review 1,

thank you very much for your valid and useful revision. Following the point by point assessment.

LINE 44- E. equinus in latin

R: Thank you, latin/italic form was updated

Line 121 - erase "clinical" specimens is enough

R: thank you, we agree with you and "clinical" has been deleted

Line 180- do not forget to consider the ocular magnification (8 or 10 ?)

R: This information has been added

Figure 1 (map) - Olbia: where is the second point?

R: The second point is near to the border on the top

Line 343- reference number is 38

R: Thank you, we correct the reference

Line 343 - the Italian experiences in this specific topic are lacking

R: Indeed the reference about the Italian experiences in this specific topic was missing and it has been added.

Line 347 - remove  the first sentence

R: deleted

To add in conclusions : More on the implementation of the role of Public Veterinary Services (AUSL) with dedicated programs on this specific context.

R: Moreover, the role of Public Veterinary Services (ASL) is a key tools to prevent the disease spread is even more crucial, as well as the implementation of specific dedicated programs and awareness.

Reviewer 2 Report

Serra et al. report results on environmental contamination by Echinococcus spp eggs as a risk for human health in Italy. The findings seem to be useful for the control of Echinococcus in infestation in the environment as well as in public.

comments:

Title- please correct as "spp." in the title.

Line 44- E. equinus should be italic.

Line 121- Please delete "clinical" since water, soil, and vegetables are not clinical samples.

Line 136,173, -  please check the consistency of "faecal or fecal".

Line 137- describe what is identified and unidentified dogs.

Line 174- please include specific gravity.

Line 256- animali- "animal?"

Line 258- please correct km2, "2 should be superscript".

Table 3- No. of sheep farm, No. of sheep

Line 308- Eggs contaminated in soil may infest to an intermediate host, not to the dog.

Line 308 and 357- is "E.g."  "E. granulosus"? please correct.

Line 344- E. granulosus should be italic.

Line 347- delete the first sentence.

Materials and Methods-

Did you record morphological features of parasite eggs after fecal examination?

Did you conduct sequencing of PCR products? please include the data.

You mentioned in the statistical analysis section that you used the chi-square test to analyze your results. However, I couldn't find any data about statistical findings such as p value, chi-square value, and so on.

If you used a chi-square test, you should identify potential risk factors for Echinococcus parasites in your study. You should also engage in discussion and conclude based on your findings.

Author Response

Dear Review2,

we would like to thank you for the time you spent to review our paper. We have addressed all your suggestions as follows:

Title- please correct as "spp." in the title.

R: The point has been added after "spp"

Line 44- E. equinus should be italic.

R: italic form has been used

Line 121- Please delete "clinical" since water, soil, and vegetables are not clinical samples.

R: we agree with you and "clinical" has been deleted

Line 136,173, -  please check the consistency of "faecal or fecal".

R: faecal has been corrected throughout the manuscript

Line 137- describe what is identified and unidentified dogs.

R: the specification has been added in the same row

Line 174- please include specific gravity.

R: density has been changed with gravity

Line 256- animali- "animal?"

R: we are sorry, it was wrong and it has been corrected

Line 258- please correct km2, "2 should be superscript".

R: we are sorry, it was a refuse corrected as Km2

Table 3- No. of sheep farm, No. of sheep

R: corrected

Line 308- Eggs contaminated in soil may infest to an intermediate host, not to the dog.

R: Indeed the sentence lacked clarity and now it reads: "The free dogs cover several kilometers and may become infected by feeding on carcasses of infested intermediate hosts abandoned in the fields. Subsequently, they may contaminate the soil with E.g. eggs potentially present in their faeces. "

Line 308 and 357- is "E.g."  "E. granulosus"? please correct.

R: corrected

Line 344- E. granulosus should be italic.

R: corrected

Line 347- delete the first sentence.

R: corrected

Materials and Methods-

Did you record morphological features of parasite eggs after fecal examination?

R: We did not record the morphological features of parasite eggs after fecal examination

Did you conduct sequencing of PCR products? please include the data.

R: In the paper we provide data on the presence of E.G. in samples collected from E.F. in Sardinia by PCR. The sequencing was not performed in this study. These data will be part of a distinct paper, which will include  a greater number of samples.

You mentioned in the statistical analysis section that you used the chi-square test to analyze your results. However, I couldn't find any data about statistical findings such as p value, chi-square value, and so on.

R: only significant p-values were reported. Following your suggestion, we included all the p-values referred to each comparison (lines 276-288)

If you used a chi-square test, you should identify potential risk factors for Echinococcus parasites in your study. You should also engage in discussion and conclude based on your findings.

R: considering that all the p-values are not statistically significant or borderline, no one risk factor can be associated with EC presence. Otherwise, following your consideration we update the discussion underlying this pint. Thank you very much
